# Radiosynthesis and Bioevaluation of ^99m^Tc-Labeled Isocyanide Ubiquicidin 29-41 Derivatives as Potential Agents for Bacterial Infection Imaging

**DOI:** 10.3390/ijms25021045

**Published:** 2024-01-15

**Authors:** Yuhao Jiang, Peiwen Han, Guangxing Yin, Qianna Wang, Junhong Feng, Qing Ruan, Di Xiao, Junbo Zhang

**Affiliations:** 1Key Laboratory of Radiopharmaceuticals of Ministry of Education, NMPA Key Laboratory for Research and Evaluation of Radiopharmaceuticals (National Medical Products Administration), College of Chemistry, Beijing Normal University, Beijing 100875, China; 202131150056@mail.bnu.edu.cn (Y.J.); 202221150118@mail.bnu.edu.cn (P.H.); 202331150067@mail.bnu.edu.cn (G.Y.); 202231150061@mail.bnu.edu.cn (Q.W.); 201921150102@mail.bnu.edu.cn (J.F.); 11132022029@bnu.edu.cn (Q.R.); 202031150054@mail.bnu.edu.cn (D.X.); 2Department of Isotopes, China Institute of Atomic Energy, P.O. Box 2108, Beijing 102413, China; 3Key Laboratory of Beam Technology of the Ministry of Education, College of Nuclear Science and Technology, Beijing Normal University, Beijing 100875, China

**Keywords:** ^99m^Tc, isocyanide, ubiquicidin 29-41, bacterial infection imaging

## Abstract

To develop a novel ^99m^Tc-labeled ubiquicidin 29-41 derivative for bacterial infection single-photon emission computed tomography (SPECT) imaging with improved target-to-nontarget ratio and lower nontarget organ uptake, a series of isocyanide ubiquicidin 29-41 derivatives (CNnUBI 29-41, *n* = 5–9) with different carbon linkers were designed, synthesized and radiolabeled with the [^99m^Tc]Tc(I)^+^ core, [^99m^Tc][Tc(I)(CO)_3_(H_2_O)_3_]^+^ core and [^99m^Tc][Tc(V)N]^2+^ core. All the complexes are hydrophilic, maintain good stability and specifically bind *Staphylococcus aureus* in vitro. The biodistribution in mice with bacterial infection and sterile inflammation demonstrated that [^99m^Tc]Tc-CN5UBI 29-41 was able to distinguish bacterial infection from sterile inflammation, which had an improved abscess uptake and a greater target-to-nontarget ratio. SPECT imaging study of [^99m^Tc]Tc-CN5UBI 29-41 in bacterial infection mice showed that there was a clear accumulation in the infection site, suggesting that this radiotracer could be a potential radiotracer for bacterial infection imaging.

## 1. Introduction

Inflammation is a complex pathophysiological response of tissues to various types of damage, such as physical, chemical, immune or microbial damage. The inflammation caused by microbial damage or the participation of microorganisms is called infection. Distinguishing infection from sterile inflammation is critical in many clinical conditions, such as endocarditis, prosthetic joint infection and spondylodiscitis [1,2]. In addition, the diagnosis and treatment of infectious diseases have received increased amounts of attention in clinical practice during and after the COVID-19 pandemic [3].

Currently, traditional methods and medical imaging tools are used to diagnose infections in clinical practice [4,5]. Traditional methods, such as microscopy based on cultures of tissue biopsies or blood samples, are invasive; however, these methods involve sampling errors and laboratory contamination and are often more suitable for diagnosis in advanced stages of infection [6,7]. Noninvasive medical imaging tools are mainly divided into anatomical structural imaging (computed tomography, magnetic resonance imaging, ultrasonography, etc.) and molecular imaging based on tissue biochemical indicators and metabolic changes [8]. As an important molecular imaging tool, nuclear medicine imaging (single-photon emission computed tomography, SPECT, and positron emission computed tomography, PET) has the characteristics of high sensitivity and good specificity and can visualize and quantitatively diagnose deep infections and track and evaluate the efficacy of treatment plans [1,9,10]. At present, nuclear medicine methods commonly used for the clinical diagnosis of infection include ^18^F-2′-fluoro-2′-deoxyglucose ([^18^F]FDG), ^18^F sodium fluoride ([^18^F]NaF), ^99m^Tc methylene diphosphonate ([^99m^Tc]Tc-MDP), in vitro labeled leukocytes ([^99m^Tc]Tc-WBCs and [^111^In]In-WBCs) and ^67^Ga gallium citrate [3,11]. These methods mainly rely on the host immune response and indirectly reflect the condition of the lesion, which also reflects the inevitable defects of this type of imaging agent depending on the immune status of the body (the number and activity of leukocytes and lymphocytes, false-negative results in immunodeficient patients) and nonspecificity (inability to distinguish infection from sterile inflammation) [11]. Nonspecific tests can easily lead to unnecessary and expensive surgeries and inevitable mortality. Therefore, pathogen-specific infection diagnosis is becoming a hot topic in nuclear medicine infection imaging [10,12].

Antimicrobial peptides (AMPs) are a class of polypeptides with antibacterial activity that are widely found in bacteria, fungi, animals, plants and even humans [13]. AMPs do not easily cause drug resistance mutations or other defects and are predicted to have broad application prospects in the pharmaceutical industry [14]. The mechanism by which AMPs exert antibacterial effects involves the electropositive groups carried by them, which specifically bind to the electronegative groups on the surface of pathogenic bacteria; thus, radiolabeled AMPs can be used for infection imaging [15].

Technetium-99m has excellent SPECT nuclide properties (E*_γ_* = 140 keV, T_1/2_ = 6.02 h) and various chemical coordination capabilities [16]. Benefiting from the success of ^99^Mo/^99m^Tc generator commercialization and the instant kit for technetium-99m radiopharmaceuticals, technetium-99m has been developed into the most widely used SPECT nuclide in clinical practice [17]. Currently, worldwide, especially in developing countries, the number of SPECT scanners is still far greater than that of PET scanners, and current trends show a powerful growth in the acceptance of state-of-the-art SPECT/CT devices equipped with cadmium-zinc-telluride (CZT) detectors, minifying multiple-pinhole collimators and (multiple-slit) slit-slat collimators [18]. Convenient sources and low prices of nuclides have greatly promoted the development of technetium-99m radiopharmaceuticals [19,20]. The development of technetium-99m radiopharmaceuticals is of great significance for the benefit of the world, especially for developing countries.

[^99m^Tc]Tc-UBI 29-41 is representative of this type of tracer. UBI 29-41 is the amino acid fragment (Figure 1) from positions 29 to 41 of the natural antimicrobial peptide ubiquicidin (UBI), and its amino acid sequence is TGRAKRRMQYNRR [21]. The mechanism by which UBI 29-41 exerts its antibacterial effect is generally believed to involve the electropositive groups (five arginines and one lysine) that it carries, which can specifically bind to the electronegative groups on the surface of pathogenic bacteria [22,23]. Since no characteristic chelating group coordinates with technetium in the molecular structure of UBI 29-41, the structure of [^99m^Tc]Tc-UBI 29-41 was speculated, and no real crystal data were acquired [24]. The introduction of chelating groups, such as the hydrazino nicotinamide ligand (HYNIC) [25,26,27], diaminedithiol ligand (N_2_S_2_) [25] and *N*-(*N*-(3-diphenylphosphinopropionyl)glycyl)-*S*-tritylcysteine ligand (PN_2_S) [28], into the N-terminal amino group and lysine amino group of UBI 29-41 molecules for technetium labeling has been reported previously. However, the reported [^99m^Tc]Tc-UBI 29-41 and its derivatives have disadvantages, such as unsuitable target/non-target ratios and high liver abdominal radioactive uptakes [29]. The introduction of suitable chelating groups and linkers coordination with technetium cores plays a significant role in regulating the in vivo pharmacokinetic properties of technetium radiopharmaceuticals.

The isocyanide group (-N≡C) is a common bifunctional linker used for ^99m^Tc labeling, which is formed when the isocyanide group coordinates with technetium, not only through a σ-coordinate bond but also through a π-feedback bond, which is a strong coordination group [30,31]. Isocyanide derivatives can coordinate different ^99m^Tc cores, such as the [^99m^Tc]Tc(I)^+^ core, [^99m^Tc][Tc(I)(CO)_3_(H_2_O)_3_]^+^ core and [^99m^Tc][Tc(V)N]^2+^ core, to form various well-stable ^99m^Tc-labeled complexes with different lipophilicities and biometabolic properties [32]. In this study, isocyanide ubiquicidin 29-41 derivatives with different carbon chain lengths were synthesized and radiolabeled with three different ^99m^Tc cores to evaluate their potential as bacterial infection SPECT imaging agents.

## 2. Results

### 2.1. Ligand Characterization

The chemical structures of isocyanides ubiquicidin 29-41 (CNnUBI 29-41, *n* = 5–9) are shown in Figure 2. The final compounds were identified by electrospray ionization mass spectrometry (ESI-MS), and the purity of the compounds was analyzed by high-performance liquid chromatography (HPLC); all the compounds were >94% pure (Appendix A).

### 2.2. Radiochemistry and Quality Control

The radiolabeling routes of ^99m^Tc-labeled CNnUBI 29-41 complexes were shown in Figure 1. [^99m^Tc]Tc-CNnUBI 29-41 complexes were obtained by a one-step and one-pot reaction with the reducing agent stannous chloride provided in a kit. [^99m^Tc]Tc(CO)_3_-CNnUBI 29-41 and [^99m^Tc]TcN-CNnUBI 29-41 complexes were obtained by a two-step reaction involving the preparation of technetium intermediates and ligand exchange reactions. First, [^99m^Tc][Tc(Ⅰ)(CO)_3_(H_2_O)_3_]^+^ and [^99m^Tc][Tc(Ⅴ)N]^2+^ intermediates were prepared as reported. Then, the optimal labeling conditions were obtained by optimizing the pH and ligand amount. In addition, the addition of Tween-80 plays a significant role in reducing the radioactive adsorption of complexes on glass bottle walls.

The radiochemical purities (RCPs) of the complexes were determined via HPLC. The HPLC results of the complexes are shown in Figure 3. [^99m^Tc]Tc-CNnUBI 29-41 and [^99m^Tc]TcN-CNnUBI 29-41 were determined by radio-HPLC system 1. The HPLC peaks of [^99m^Tc]Tc-CNnUBI 29-41 were all double peaks and were mainly in the range of 8 to 10 min (Appendix A and Appendix A). The retention times of [^99m^Tc]TcN-CNnUBI 29-41 (*n* = 5–9) were approximately 10 min (Appendix A and Appendix A). Under the same conditions, the retention times of [^99m^Tc]TcO_4_^−^ and [^99m^Tc][TcN]^2+^ were 3.918 min and 2.528 min, respectively (Appendix A). [^99m^Tc]Tc(CO)_3_-CNnUBI 29-41 was determined by radio-HPLC system 2. The retention times of [^99m^Tc]Tc(CO)_3_-CNnUBI 29-41 (*n* = 5–9) were 15.308, 15.583, 15.942, 16.731 and 17.038 min, respectively (Appendix A and Appendix A). Under the same conditions, the retention times of [^99m^Tc]TcO_4_^−^ and [^99m^Tc][Tc(CO)_3_(H_2_O)_3_]^+^ were 3.918 min and 12.588 min, respectively (Appendix A). The HPLC analysis results showed that the RCPs of all the complexes were greater than 90%. No additional purification was carried out for further evaluation of any of the complexes.

### 2.3. In Vitro Physicochemical Characterization

#### 2.3.1. In Vitro Stability Tests

[^99m^Tc]Tc-CNnUBI 29-41 was stable in saline and mouse serum for 6 h, as shown in the HPLC patterns (Appendix A). [^99m^Tc]Tc(CO)_3_-CNnUBI 29-41 and [^99m^Tc]TcN-CNnUBI 29-41 were stable in saline and mouse serum, respectively, for 4 h, as shown in the HPLC diagrams (Appendix A, respectively). There were more than 90% RCPs for the radiotracers under these conditions, suggesting that the complexes have good stability in vitro.

#### 2.3.2. Determination of the Distribution Coefficients of the Complexes

The distribution coefficient (log D) values of [^99m^Tc]Tc-CNnUBI 29-41, [^99m^Tc]Tc(CO)_3_-CNnUBI 29-41 and [^99m^Tc]TcN-CNnUBI 29-41 are shown in Table 1. The results showed that all the tracers were hydrophilic. Moreover, when the ^99m^Tc-labeling core was [^99m^Tc][Tc(CO)_3_(H_2_O)_3_]^+^, the complexes were more lipophilic than the other two series of complexes. The lipophilicities of the [^99m^Tc][Tc(Ⅰ)]^+^ core complexes were similar to those of the [^99m^Tc][Tc(Ⅴ)N]^2+^ core complexes with the same radioligands. For the same ^99m^Tc-labeled core, with increasing carbon chain number, the value of log D increases, which means that the lipophilicity increases.

### 2.4. In Vitro Binding of Complexes to Bacteria

The results of the in vitro bacterial binding studies of [^99m^Tc]Tc-CNnUBI 29-41, [^99m^Tc]Tc(CO)_3_-CNnUBI 29-41 and [^99m^Tc]TcN-CNnUBI 29-41 are shown in Figure 4. Excess UBI 29-41 was added for competition. The binding values of the radiotracers to bacteria were set to 100% as control groups, and the data for the binding of the complexes to the bacteria competing with UBI 29-41 are shown as the binding value/control group ratio. The bacterial binding rates of [^99m^Tc]Tc-CNnUBI 29-41 (*n* = 5–9) decreased by 40.56%, 32.18%, 34.21%, 21.41% and 26.96%, respectively. The bacterial binding rates of [^99m^Tc]Tc(CO)_3_-CNnUBI 29-41 (*n* = 5–9)decreased by 32.89%, 23.42%, 28.83%, 31.15% and 12.92%, respectively. The bacterial binding rates of [^99m^Tc]TcN-CNnUBI 29-41 (*n* = 5–9) decreased by 31.19%, 29.58%, 22.77%, 22.84% and 15.96%, respectively. Among the ^99m^Tc-labeled complexes tested, CN5UBI 29-41 had the highest inhibitory effect on the ligand, while CN9UBI 29-41 had the lowest inhibitory effect. Overall, as the length of the carbon chain of the isonitrile ligand increases, the inhibition rate of the ^99m^Tc-labeled complexes gradually decreases. These results indicated that the binding of the complexes to the bacteria was significantly reduced after the inhibitor UBI 29-41 was added and that all the complexes specifically bound to the bacteria.

### 2.5. Biodistribution

The biodistribution data for [^99m^Tc]Tc-CNnUBI 29-41, [^99m^Tc]Tc(CO)_3_-CNnUBI 29-41 and [^99m^Tc]TcN-CNnUBI 29-41 from mice bearing bacterial infections are shown in Appendix A and Figure 5. Comparisons of abscess-to-muscle and abscess-to-blood ratios for different ^99m^Tc-labeled complexes are shown in Figure 6. The biodistribution data of [^99m^Tc]Tc-CN5UBI 29-41 at different postinjection times are shown in Appendix A and Figure 7.

As shown in Figure 5a and Appendix A, in the bacterial infection mouse models, the abscess uptake values of [^99m^Tc]Tc-CN5UBI 29-41 (1.44 ± 0.46%ID/g) and [^99m^Tc]Tc-CN6UBI 29-41 (1.70 ± 0.23%ID/g) were greater than those of [^99m^Tc]Tc-CNnUBI 29-41 (*n* = 7–9), while the abscess uptake values of [^99m^Tc]Tc-CNnUBI 29-41 (*n* = 7–9) were not significantly different. The abscess-to-muscle ratio of [^99m^Tc]Tc-CN5UBI 29-41 (4.15 ± 0.49) was the highest of the five [^99m^Tc]Tc-CNnUBI 29-41 complexes. The abscess-to-blood ratios of the [^99m^Tc]Tc-CNnUBI 29-41 complexes were similar. The tracers [^99m^Tc]Tc-CNnUBI 29-41 are metabolized mainly by the kidneys and liver. As the carbon chain length of the ligand increases, the lipophilicity of the complex increases. As a result of biodistribution, the uptake in the liver and intestines gradually increases.

As shown in Figure 5b and Appendix A, in the bacterial infection mouse models, the abscess uptake values of [^99m^Tc]Tc(CO)_3_-CN5UBI 29-41 (3.44 ± 0.89%ID/g) were the highest among [^99m^Tc]Tc(CO)_3_-CNnUBI 29-41 (*n* = 5–9), while the muscle and blood uptake values were also the highest. The abscess-to-muscle and abscess-to-blood ratios of the [^99m^Tc]Tc(CO)_3_-CNnUBI 29-41 complexes were similar. The tracers [^99m^Tc]Tc(CO)_3_-CNnUBI 29-41 are metabolized mainly by the kidneys and liver. As the carbon chain length of the ligand increases, the lipophilicity of the complex increases. As a result of biodistribution, the uptakes of the liver and spleen gradually increase, and the proportion of liver metabolism gradually increases. Among the three series of complexes, the uptake of [^99m^Tc]Tc(CO)_3_-CNnUBI 29-41 complexes by organs and tissues was generally greater than that of the other two series of complexes.

As shown in Figure 5c and Appendix A, in the bacterial infection mouse models, the abscess uptake of [^99m^Tc]TcN-CN5UBI 29-41 (0.18 ± 0.05%ID/g) was the lowest among the [^99m^Tc]TcN-CNnUBI 29-41 complexes (*n* = 5–9), while the abscess uptakes of [^99m^Tc]TcN-CNnUBI 29-41 (*n* = 6–9) were not significantly different. The abscess-to-muscle and abscess-to-blood ratios of the [^99m^Tc]TcN-CNnUBI 29-41 complexes were similar. The tracers [^99m^Tc]TcN-CNnUBI 29-41 are metabolized mainly by the kidneys and liver. As the carbon chain length of the ligand increases, the uptake in the liver gradually increases, and the uptake in the kidneys gradually decreases. Regarding the other organs, the low radioactivity uptake in the stomach and thyroid indicated that the three series of complexes had good stability in vivo. Among the three series of complexes, the uptake of [^99m^Tc]TcN-CNnUBI 29-41 complexes by organs and tissues was generally lower than that of the other two series of complexes.

Figure 6 shows that all abscess-to-blood ratios in the three series of ^99m^Tc-labeled CNnUBI 29-41 complexes were close and less than 1. The abscess-to-muscle ratios ranged from 2.10 ± 0.80 to 4.15 ± 0.49. Among them, the abscess-to-muscle ratio was highest for [^99m^Tc]Tc-CN5UBI 29-41. Therefore, additional investigations of the effects of injection time and mouse model of inflammation were carried out on [^99m^Tc]Tc-CN5UBI 29-41.

From the biodistribution results of [^99m^Tc]Tc-CN5UBI 29-41 at different injection times shown in Figure 7a and Appendix A, good abscess uptake was observed from 30 min (1.64 ± 0.40%ID/g) to 240 min (1.42 ± 0.59%ID/g), and it also remained for 240 min. From 30 to 120 min, the uptake in muscle and blood decreased by approximately half, while from 120 to 240 min, there was little change in the uptake in muscle and blood. Therefore, from 30 to 120 min, the abscess-to-muscle and abscess-to-blood ratios in bacterial infection mice increased, and the ratios did not change from 120 to 240 min. [^99m^Tc]Tc-CN5UBI 29-41 has high blood retention.

The biodistribution data of [^99m^Tc]Tc-CN5UBI 29-41 in mice bearing turpentine-induced abscesses are shown in Figure 7a and Appendix A. Comparisons of abscess-to-muscle and abscess-to-blood ratios during bacterial infection and sterile inflammation in [^99m^Tc]Tc-CN5UBI 29-41 are shown in Figure 7b and Appendix A. The abscess uptake of the bacterium-infected mice (1.44 ± 0.48%ID/g) was significantly greater than that of the sterile inflammation mice (0.73 ± 0.05%ID/g) at 120 min postinjection. The abscess-to-muscle and abscess-to-blood ratios (4.15 ± 0.49 and 0.81 ± 0.05) of the bacterial infection mice were also definitely greater than those of the sterile inflammation mice (2.52 ± 0.22 and 0.59 ± 0.06) at 120 min postinjection. These results indicated that [^99m^Tc]Tc-CN5UBI 29-41 can distinguish between bacterial infection and sterile inflammation.

### 2.6. SPECT Imaging

In the SPECT imaging studies, 30, 120 and 240 min were chosen for SPECT imaging. As shown in Figure 8, [^99m^Tc]Tc-CN5UBI 29-41 accumulated at the site of bacterial infection in the left forelimb of the mouse. SPECT images were also consistent with the biodistribution results. The kidneys, liver, intestines, gallbladder and cardiac blood pool could be clearly visualized via SPECT images. There was no radioactivity accumulation in the thyroid or stomach of the mice, suggesting that [^99m^Tc]Tc-CN5UBI 29-41 exhibits good stability in mice in vivo. As time elapses, the radioactivity accumulation in the mouse cardiac blood pool gradually decreases, while the radioactivity accumulation in the infection site still remains stable. On SPECT images at different time points, it can be observed that the contrast of the abscess site in the bacterial infection mice increased with time.

## 3. Discussion

UBI 29-41 is considered a potential target molecule for bacterial infections. [^99m^Tc]Tc-UBI 29-41 has been reported for more than 20 years. Its instant kit has also been successfully prepared [21,33,34]. A large number of clinical trials have shown its excellent properties in detecting bacterial infections, especially prosthetic joint infection [35], spondylodiscitis [36], mediastinal infection after cardiac surgery [37], etc. However, currently, there is no chelating group on the UBI 29-41 molecule that can directly coordinate with ^99m^Tc, and the molecular structure of [^99m^Tc]Tc-UBI 29-41 is speculative and unclear [24]. There are currently a large number of new UBI 29-41 derivatives for bacterial infection imaging, including ^99m^Tc-labeling, ^68^Ga-labeling, and fluorescent molecular probes [38]. Most UBI 29-41 derivatives are directly modified on the *N*-terminal amino group of the polypeptide chain. According to the structural data of [^99m^Tc]Tc-UBI 29-41, the amino group of lysine and the guanidine group of the adjacent arginine coordinate with the ^99m^Tc-oxo core [39]. This indicates that the amino group of lysine in the UBI 29-41 molecule can be modified. In addition, the activity of the lysine amino group is also greater than that of the *N*-terminal amino group. In 2015, Yueqing Gu et al. reported that the fluorescent group ICG02, which is modified on the lysine amino group of the UBI 29-41 molecule (ICG02-UBI 29-41), can locate bacterial infections [40]. The modification of the lysine amino group in the UBI 29-41 molecule introduces different carbon chain lengths of the isocyano group, which coordinates with ^99m^Tc. This could clarify the structure of the ^99m^Tc isonitrile complex. The guanidine group can be released and was originally coordinated with technetium in the [^99m^Tc]Tc-UBI 29-41 complex. The release of the guanidine group improved the cation degree of the labeled peptides and their ability to bind to bacterial surfaces.

Isocyanide derivatives are monodentate ligands that can coordinate different ^99m^Tc cores, such as the [^99m^Tc]Tc(I)^+^ core, [^99m^Tc][Tc(I)(CO)_3_(H_2_O)_3_]^+^ core and [^99m^Tc][Tc(Ⅴ)N]^2+^ core, to form various highly stable ^99m^Tc-labeled complexes with different lipophilicities and biometabolic properties. For example, it is common to form a six-coordinate system with a [^99m^Tc]Tc(I)^+^ core, a three-coordinate system with a [^99m^Tc] [Tc(I)(CO)_3_(H_2_O)_3_]^+^ core and [^99m^Tc][Tc(V)N]^2+^ core, and a ‘4 + 1’ complex with NS_3_ [41]. Here, we selected three types of ^99m^Tc cores for labeling, namely, the [^99m^Tc]Tc(I)^+^ core, [^99m^Tc] [Tc(I)(CO)_3_(H_2_O)_3_]^+^ core and [^99m^Tc][Tc(V)N]^2+^ core. The HPLC peaks of [^99m^Tc]Tc-CNnUBI 29-41 were all double peaks, which may be related to the structure of the complexes. Due to the large size of CNnUBI 29-41 molecules, there are structural molecules that do not fully form six coordination when coordinating with technetium, resulting in the formation of multiple ligand molecules with different coordination numbers in the final complexes. The UBI 29-41 molecule contains several guanidine, carboxyl and amino groups. These peptides with multiple electrophilic groups and ionizing groups exhibit varying degrees of protonation in the slightly acidic HPLC mobile phase, resulting in a certain equilibrium relationship between the binding electron sites of the complex [42]. Therefore, broad double peaks appear in the HPLC diagram. This can also be attributed to the wider peaks in the HPLC chromatograms of the [^99m^Tc]Tc(CO)_3_-CNnUBI 29-41 complexes. The structure of the ^99m^Tc-nitrido-labeled isonitrile complex is speculated to be the six-coordinate distorted octahedral structure reported in the reference [43]. During the labeling process, the ligands [^99m^Tc]Tc-CNnUBI 29-41 were the least common among the three technetium cores, which is closely related to the structure and labeling process of the labeling complexes. Direct labeling was performed via a one-step reaction method involving direct coordination with the isonitrile group after the reduction of sodium pertechnetate. This reaction process involves both reduction and coordination. The tricarbonyl technetium-labeled isonitrile complex undergoes a two-step separate reaction, first by reducing sodium pertechnetate and coordinating with CO to obtain the [^99m^Tc][Tc(I)(CO)_3_(H_2_O)_3_]^+^ intermediate. Then, a ligand exchange reaction occurs between the isonitrile ligand and the intermediate. The second step of the ligand exchange reaction requires consideration of the effects of the pH and ligand amount. Generally, the larger the ligand amount, the more complete the ligand exchange under suitable pH conditions. The number of ligands used in the ^99m^Tc-nitrido-labeled isonitrile complex is the highest, which is closely related to the molecules displaced by the second step of the ligand exchange reaction. The stronger the coordination ability of the displaced molecules, the more ligands are needed. 1,3-Propylenediaminetetraacetic acid (PDTA), which has a stronger ability to bind technetium than H_2_O, was added to this SDH kit. Therefore, additional isonitrile ligands need to be added when preparing ^99m^Tc-nitrido-labeled isonitrile complexes.

The surfactant polyoxyethylene-80-sorbitan monooleate (Tween-80) needs to be added during the radiolabeling process of CNnUBI 29-41. If Tween-80 is not added before the labeling reaction, most of the generated radiolabeling products will be adsorbed on the wall of the glass penicillin bottle. In addition, the addition of Tween-80 after the labeling reaction prevents the removal of the radiolabeled complex from the bottle wall. After adding Tween-80 before the radiolabeling reaction, most of the products were in solution and had little adsorption on the glass bottle wall, which made subsequent research easier.

The radiochemical purities (RCPs) of all the ^99m^Tc-labeled complexes were greater than 90%, and these complexes were used directly for subsequent research without further purification. According to the in vitro stability experiments, more than 90% of the RCPs of the complexes were stable, indicating that the complexes had good stability. On the basis of the distribution coefficient results, three series of ^99m^Tc-labeled isonitrile complexes are hydrophilic (log *D* < 0). For the same isonitrile ligand, among the three series of technetium-labeled cores, the ^99m^Tc(CO)_3_-labeled complex was the most lipophilic, which could be related to the obvious lipophilic properties of the tricarbonyl technetium core itself. In addition, for each type of technetium core, the lipophilicity of the ^99m^Tc-labeled complexes gradually increases with increasing carbon chain number of the isonitrile ligand. In the in vitro bacterial binding assay, the competition results showed that the bacterial binding efficiency of the blocked group was significantly lower than that of the control group, which showed that the specific binding of the complexes to bacteria was inhibited by UBI 29-41. The inhibition rate of UBI 29-41 decreased with increasing carbon chain length while the nonspecific binding of the complexes increased. This difference may be related to the increased lipophilicity of the complexes.

The biodistribution of 15 complexes was studied in bacterium-infected mice at 120 min postinjection. The biodistribution of the complexes was closely related to the type of technetium core, lipophilicity and number of carbon chains on the isonitrile ligand. Similarly, the muscle uptake of the [^99m^Tc]Tc-CNnUBI 29-41 complex was similar at 120 min postinjection. The abscess uptakes of [^99m^Tc]Tc-CN5UBI 29-41 and [^99m^Tc]Tc-CN6UBI 29-41 were greater than that of the other [^99m^Tc]Tc-CNnUBI 29-41 complexes. The target-to-nontarget ratio of [^99m^Tc]Tc-CN5UBI 29-41 was the highest. Overall, the biodistribution of the [^99m^Tc]Tc-CNnUBI 29-41 complexes was optimal because of their suitable lipophilicity and high affinity for multiple coordinating molecules. The uptake of the [^99m^Tc]Tc(CO)_3_-CNnUBI 29-41 complex was greater than that of the other two complexes in all organs and tissues, with significantly greater uptake in the heart, liver, spleen, intestine and blood. The target-to-nontarget ratios of the [^99m^Tc]Tc(CO)_3_-CNnUBI 29-41 complexes were not high; they were all between 2.42 ± 0.37 and 2.60 ± 0.68. Significant uptake was observed in the liver and abdomen via SPECT (Appendix A). The uptake of the [^99m^Tc]TcN-CNnUBI 29-41 complexes in all organs and tissues was relatively low, indicating that these tracers are metabolized faster in organisms. Although the target-to-nontarget ratios of the [^99m^Tc]TcN-CNnUBI 29-41 complexes were between 2.10 ± 0.80 and 3.32 ± 0.99, the absolute uptake of abscess was less than 0.5%ID/g. The bacterial infection site could be almost imperceptible in the SPECT image (Appendix A). The organ uptakes of the [^99m^Tc]Tc-CNnUBI 29-41 complexes and [^99m^Tc]Tc(CO)_3_-CNnUBI 29-41 complexes at the bacterial infection site were significantly greater than that of [^99m^Tc]TcN-CNnUBI 29-41, which may be related to the overall charge properties of the complex and the valence of technetium. The [^99m^Tc]Tc-CNnUBI 29-41 complexes and [^99m^Tc]Tc(CO)_3_-CNnUBI 29-41 complexes had a +1 valence charge overall [31], while the [^99m^Tc]TcN-CNnUBI 29-41 complexes were electrically neutral [43]. Molecules with positive charges are more likely to bind to negatively charged bacterial surfaces. Secondly, the valence of technetium in the [^99m^Tc]Tc-CNnUBI 29-41 complexes and [^99m^Tc]Tc(CO)_3_-CNnUBI 29-41 complexes is +1 [31], while the valence of technetium in the [^99m^Tc]TcN-CNnUBI 29-41 complexes is +5 [32,43]. The valence of technetium in the technetium complex may also have a significant impact on the biodistribution properties and images. For example, [^99m^Tc]Tc(V)-dimercaptosuccinic acid (DMSA) labeled under alkaline conditions is used as a soft tissue tumor imaging agent [44], while [^99m^Tc]Tc(III)-DMSA labeled under acidic conditions is used for renal cortical imaging [45,46,47,48].

Considering the biodistribution, abscess uptake, target-to-nontarget ratio and nontarget organ uptake of each complex in bacterium-infected mice, we ultimately chose [^99m^Tc]Tc-CN5UBI 29-41 for further research. Biodistribution of [^99m^Tc]Tc-CN5UBI 29-41 was carried out at 30, 120 and 240 min postinjection. The uptake in most organs decreased from 30 to 120 min. Within 120 to 240 min, there was a slight decrease in blood uptake, with less significant changes in the uptake of other organs. The above changes can also be observed via SPECT imaging. A control experiment was conducted on a sterile inflammation mouse model. At 120 min postinjection, the uptake at the abscess site (1.44 ± 0.48%ID/g) in the bacterial-infected mice was significantly greater than that in the sterile inflammation mice (0.73 ± 0.05%ID/g). A comparison of the target-to-nontarget ratios of the two mouse models revealed that the target-to-nontarget ratio of the bacterial infection sites was significantly greater than that of the sterile inflammation sites. These findings indicate that [^99m^Tc]Tc-CN5UBI 29-41 can distinguish between bacterial infection and sterile inflammation. SPECT imaging was performed on bacterium-infected mice at 30, 120 and 240 min postinjection. Significant radioactive accumulation was observed in the infected sites of the left upper limb of the mouse at 30 to 240 min postinjection via SPECT images. The development of the cardiac blood pool gradually decreased over time, indicating that the radioactivity in the blood was gradually cleared. The gallbladder, kidneys and intestines in nontarget organs exhibited significant uptake, with slight uptake by the liver, further indicating that the tracer is metabolized and excreted mainly by the kidneys, followed by hepatic metabolism and intestinal excretion. The SPECT imaging results were also consistent with the biodistribution results in Appendix A.

Compared with the biodistribution results of [^99m^Tc]Tc-UBI 29-41 in the literature, the target-to-nontarget ratios of [^99m^Tc]Tc-UBI 29-41 at bacterial infection sites were 1.7–4.7 [9,21,23,25,49,50,51], which are close to the target-to-nontarget ratio of [^99m^Tc]Tc-CN5UBI 29-41 (4.15 ± 0.49 at 120 min postinjection). The target-to-nontarget ratios at sterile inflammation sites in [^99m^Tc]Tc-UBI 29-41 were 1.1–1.2. Compared to that in [^99m^Tc]Tc-UBI 29-41, the target-to-nontarget ratio in sterile inflammation (2.52 ± 0.22 at 120 min postinjection) in [^99m^Tc]Tc-CN5UBI 29-41 was greater, which may be related to partial nonspecific uptake.

Nevertheless, the biodistribution and SPECT imaging results still need to be improved. Due to the carbon chain connecting UBI 29-41 to the isonitrile group, the ^99m^Tc-labeled isonitrile complex has a certain degree of lipophilicity. Changing the carbon chain to a PEG chain in isocyanide UBI 29-41 derivatives may further reduce the nontarget uptake of the complex and increase the target-to-nontarget ratio. In addition, three series of ^99m^Tc-labeled CNnUBI 29-41 complexes are multicoordinate, with multiple targeted structures and larger molecules. The multicoordinate complexes had longer retention times in the blood, and the target-to-blood ratios of the three types of complexes were less than 1. Mixed monodentate isonitrile complexes, such as “4 + 1” tris(2-mercaptoethyl)-amine–isocyanide complexes [41,52] and “2 + 1” dithiocarbamate/acetylacetonate–isocyanide complexes [53,54,55], can be designed. Through such a design, the coordination number is reduced, and the metabolic rate of blood and the target-to-blood ratio can be improved. In addition, the nontarget uptake decreased. Moreover, the coordination chemistry of technetium is rich and diverse. Different bifunctional chelating groups can be introduced into the molecular design of UBI 29-41, such as N_2_S_2_ diamidedithiols (DADT) [56], monoamine-monoamide dithiol (MAMA) [57], small peptides Gly-Ala-Gly-Gly and Gly-Ser-Cys [58]. The chemical valence of technetium ranges from −1 to +7. There are various technetium cores, such as the [^99m^Tc][Tc(CO)_3_]^+^ core, [^99m^Tc][TcO]^3+^ core, [^99m^Tc][TcN]^2+^ core and [^99m^Tc][TcO_2_]^+^ core [20,58]. In addition, the selection of linker types and lengths between suitable targeting and chelating groups is important and has a significant influence on regulating pharmacokinetic and targeting properties.

## 4. Materials and Methods

### 4.1. Materials

CNnUBI 29-41 was synthesized and purchased from Suzhou Tianma Pharmaceutical Group Tianji Biopharmaceutical Co., Ltd., Suzhou, China. All the other reagents were used without further purification. [^99m^Tc]NaTcO_4_ was obtained from a ^99^Mo/^99m^Tc generator purchased from Atomic High Tech Co., Ltd., Beijing, China. A blank kit, which contains 2.6 mg of sodium citrate, 1 mg of *_L_*-cysteine, 60 μg of stannous chloride and 10 mg of mannitol, and SDH kit, which contains 0.05 mg of stannous chloride dihydrate, 5.0 mg of SDH and 5.0 mg of propylenediamine tetraacetic acid (PDTA), were obtained from Beijing Shihong Pharmaceutical Co., Ltd., Beijing, China. HPLC analysis was performed using a Shimadzu UV system and a Gabi raytest radioactivity detector with an analytical column (C-18, 100–5 μm, 250 × 4.6 mm, Kromasil, Eka Chemicals, Bohus, Sweden). The centrifuge model was 800-1 (Ronghua, Jintan, China). The radioactivity was measured by a WIZARD2 2480 Automatic Gamma Counter (PerkinElmer, Waltham, MA, USA), an HRS-2000 technetium analyzer (Huaruisen, Beijing, China) and an RW-905A radioactivity meter (Hechang, Beijing, China). A SPECT/CT imaging study was carried out on a Triumph SPECT/CT scanner (TriFoil Imaging, Los Angeles, CA, USA). Female Kunming mice (18–22 g) were purchased from Beijing Vital River Laboratory Animal Technology, Beijing, China.

### 4.2. Ligand Characterization

The chemical formula, HPLC purity and MS analysis of the ligand CNnUBI 29-41 (*n* = 5–9) are shown in Table 2.

### 4.3. Radiochemistry and Quality Control

[^99m^Tc]Tc-CNnUBI 29-41: A blank kit was dissolved in 300 μL of saline. Then, 50 μL of CNnUBI 29-41 (1 mg/mL, H_2_O), 100 μL of Tween-80 (100 mg/mL, saline) and 500–600 μL of freshly eluted [^99m^Tc]NaTcO_4_ solution (74–740 MBq) were sequentially added to the solution. The mixture was shaken and heated at 100 °C for 30 min and cooled to room temperature.

[^99m^Tc]Tc(CO)_3_-CNnUBI 29-41: Step 1: Preparation of [^99m^Tc][Tc(CO)_3_(H_2_O)_3_]^+^ intermediate: In a 10 mL penicillin vial, 15 mg of potassium sodium tartrate, 10 mg of sodium borohydride and 5 mg of sodium carbonate were accurately weighed and dissolved in 0.5 mL of saline to completely dissolve. After sealing, CO gas was continuously introduced into the mixed solution for 15 min. Then, 1 mL of freshly eluted [^99m^Tc]NaTcO_4_ solution (74–740 MBq) was added to the solution. The mixture was shaken and heated at 80 °C with CO for 30 min. After the reaction, the final solution was cooled to room temperature, after which the radiochemical purity was determined via HPLC. Step 2: Preparation of [^99m^Tc]Tc(CO)_3_-CNnUBI 29-41: The pH of the intermediate solution was adjusted to 8–9 by using 1 N HCl. Then, 300 μL of CNnUBI 29-41 (0.5 mg/mL, H_2_O), 100 μL of Tween-80 (5 mg/mL, saline) and 500–600 μL of [^99m^Tc][Tc(CO)_3_(H_2_O)_3_]^+^ intermediate solution (74–740 MBq) were mixed and added to a 10 mL vacuum penicillin vial. The mixture was reacted at 100 °C for 30 min and cooled to room temperature.

[^99m^Tc]TcN-CNnUBI 29-41: Step 1: Preparation of the [^99m^Tc][TcN]^2+^ intermediate: One milliliter of freshly eluted [^99m^Tc]NaTcO_4_ solution (37–740 MBq) was added to the SDH kit. After the mixture was fully shaken and the freeze-dried substance in the SDH kit was completely dissolved, the mixture was incubated at room temperature for 15 min. The radiochemical purity was subsequently determined via HPLC. Step 2: Preparation of [^99m^Tc]TcN-CNnUBI 29-41: 300 μL of CNnUBI 29-41 (2 mg/mL, H_2_O), 100 μL of Tween-80 (100 mg/mL, saline) and 500–600 μL of [^99m^Tc][TcN]^2+^ intermediate solution (74–740 MBq) were mixed and added to a 10 mL vacuum penicillin vial. The mixture was reacted at 100 °C for 30 min and cooled to room temperature.

The radiochemical purities of the radiolabeled complexes were determined by high-performance liquid chromatography (HPLC). The HPLC analysis was performed under the following two conditions. Two analytical HPLC procedures were carried out on the same analytical column at a flow rate of 1.0 mL/min as follows: System 1: (solvent A: purified water with 0.1% trifluoroacetic acid, solvent B: acetonitrile with 0.1% trifluoroacetic acid) 0–2 min, 10% B; 2–5 min, 10–90% B; 5–18 min, 90% B; and 18–25 min, 90–10% B; and System 2: (solvent A: purified water with 0.1% trifluoroacetic acid, solvent B: methanol with 0.1% trifluoroacetic acid) 0–2 min, 10% B; 2–15 min, 10–90% B; 15–20 min, 90% B; and 20–25 min, 90–10% B. The radiochemical purities of [^99m^Tc]Tc-CNnUIBI 29-41, [^99m^Tc][TcN]^2+^ intermediate and [^99m^Tc]TcN-CNnUBI 29-41 were determined by radio-HPLC System 1. The radiochemical purities of the [^99m^Tc][Tc(CO)_3_(H_2_O)_3_]^+^ intermediate and [^99m^Tc]Tc(CO)_3_-CNnUBI 29-41 were determined by radio-HPLC System 2.

### 4.4. In Vitro Physicochemical Characterization

#### 4.4.1. In Vitro Stability Study

The in vitro stabilities of [^99m^Tc]Tc-CNnUBI 29-41 were evaluated by measuring the RCP of each complex after remaining in saline at room temperature for 6 h and after treatment in mouse serum at 37 °C for 6 h. The in vitro stabilities of [^99m^Tc]Tc(CO)_3_-CNnUBI 29-41 and [^99m^Tc]TcN-CNnUBI 29-41 were evaluated by measuring the RCP of each complex after remaining in saline at room temperature for 4 h and after treatment in mouse serum at 37 °C for 4 h.

#### 4.4.2. Distribution Coefficient (Log D) Measurements

The lipophilicities of the ^99m^Tc-labeled complexes were evaluated by the distribution coefficient. The method for determining the distribution coefficient was as follows: 0.1 mL of the radiolabeled complex (3.7 MBq) was mixed with 0.9 mL of phosphate buffer (0.025 mol/L, pH 7.4) and 1 mL of *n*-octanol in a 5 mL centrifuge tube. The mixture was shaken on a vortex mixer at room temperature for 3 min and centrifuged at 5000 r/min for 5 min. Aliquots of 0.1 mL of both *n*-octanol and PBS were collected, and the radioactivity was determined with a γ-counter. The distribution coefficient was calculated as follows: Log *D* = lg[(counts per minute in *n*-octanol/counts per minute in PBS)]. Each group of measurements was carried out five times in parallel. The final results are expressed as the log D ± SD.

### 4.5. In Vitro Bacterial Binding Experiments

The in vitro bacterial binding of the ^99m^Tc-labeled complexes was evaluated by using a previously reported method [23]. Generally, 0.1 mL of 370 kBq ^99m^Tc-labeled complex solution and 0.1 mL of sodium phosphate buffer (PBS) containing approximately 1 × 10^8^ *S. aureus* were added to a test tube with 0.8 mL of the incubation buffer (0.05 M PBS containing 0.1% Tween-80 and 0.2% acetic acid, pH = 5). The mixture was mixed by vortexing and incubated for 1 h at 37 °C, after which the tubes were centrifuged at 12,000 r/min for 5 min. Afterwards, the pellets were washed and resuspended in 1 mL of incubation buffer. The mixture was subsequently recentrifuged as described above. The removed supernatant and bacterial pellets were collected, and the radioactivity was measured with a γ-counter. To determine the specificity of binding to bacteria, an in vitro competition study was carried out. In competition experiments, the bacteria were preincubated with 100-fold excess UBI 29-41 for 1 h at 37 °C. Then, the ^99m^Tc-labeled complexes were added, followed by incubation for 1 h at 37 °C. The bacterial binding value was calculated as described above. The total tube was incubated without bacteria. Each group of measurements was carried out five times in parallel. The results are expressed as the mean ± SD.

### 4.6. Biodistribution Studies in Mice with Bacterial Infection

Bacterial infection mouse models were constructed as follows: *S. aureus* (ATCC 25923) was cultured in a liquid medium at 37 °C for approximately 24 h, washed with PBS (pH 7.4) and diluted to obtain a suspension of 1 × 10^8^ CFU/mL. A 0.1 mL suspension of *S. aureus* was injected into the left thigh muscle of female Kunming mice (18–22 g). After 24 h, obvious swelling was observed in the left thigh muscle of the mice. Each mouse was injected with 0.1 mL of ^99m^Tc-labeled complexes (3.7 MBq/mL) via the tail vein. The mice were sacrificed with isoflurane anesthesia at 120 min postinjection. The organs and tissues of interest (the infected muscle in the left thigh, normal muscle in the right thigh, blood, heart, liver, lung, kidney, spleen, stomach, bone, small intestine and large intestine) were collected, weighed and measured by a γ-counter. The results are expressed as the percentage of the injected dose per gram of tissue ± standard deviation (%ID/g ± SD). After the mice were sacrificed, the tissues containing the thyroid around the trachea in the neck were collected and measured. The results are expressed as the percentage of the injected dose ± standard deviation (%ID ± SD). To evaluate the metabolic properties of [^99m^Tc]Tc-CN5UBI 29-41, biodistribution studies were carried out in mice infected with *S. aureus* at 30, 120 and 240 min.

### 4.7. Biodistribution Studies in Mice with Turpentine-Induced Abscesses

The sterile inflammation mouse model was constructed as follows: turpentine oil (0.1 mL) was injected into the left thigh muscle of female Kunming mice (18–22 g). After 48 h, obvious swelling was observed in the left thigh muscle of the mice. Each mouse was injected with 0.1 mL of [^99m^Tc]Tc-CN5UBI 29-41 (3.7 MBq/mL) via the tail vein. The mice were sacrificed under anesthesia with 1.5% isoflurane at 120 min postinjection. The organs of interest (the inflamed muscle in the left thigh, normal muscle in the right thigh, blood, heart, liver, lung, kidney, spleen, stomach, bone, small intestine and large intestine) were collected, weighed and measured by a γ-counter. The results are expressed as the percentage of the injected dose per gram of tissue ± standard deviation (%ID/g ± SD). After the mice were sacrificed, the tissues containing the thyroid around the trachea in the neck were collected and measured. The results are expressed as the percentage of the injected dose ± standard deviation (%ID ± SD).

### 4.8. SPECT Imaging Study

The bacterial-infected mouse model was generated as follows: 100 μL of bacterial suspension containing approximately 1 × 10^8^ CFU/mL (for *S. aureus*) was injected into the left forelimb muscle of the same mice. Approximately 24 h later, obvious swelling was observed in the left forelimb muscle of the mice. One hundred microliters of [^99m^Tc]Tc-CN5UBI 29-41 (30~37 MBq) was injected intravenously into each bacterially infected mouse via the tail vein (*n* = 2). A SPECT/CT imaging study was performed with bacterial-infected mice anesthetized with 1.5% isoflurane at 120 min postinjection. The SPECT/CT images were reconstructed with HiSPECT 3.2.0 software (Bioscan, Washington, DC, USA) and processed with VivoQuant 2.5 software (Invicro, Needham, MA, USA).

### 4.9. Statistical and Data Analyses

All the statistical analyses were performed with Microsoft Office Excel 2019. The quantitative data are presented as the mean ± standard deviation. The data were analyzed using an unpaired two-tailed Student’s *t*-test. *p* < 0.05 indicated statistical significance.

## 5. Conclusions

In this work, five isocyanide ubiquicidin 29-41 derivatives (CNnUBI 29-41, *n* = 5–9) were successfully prepared and radiolabeled with three different ^99m^Tc cores ([^99m^Tc]Tc(I)^+^ core, [^99m^Tc][Tc(I)(CO)_3_(H_2_O)_3_]^+^ core and [^99m^Tc][Tc(V)N]^2+^ core) to obtain fifteen ^99m^Tc-labeled complexes in high radiochemical yield. All the complexes were hydrophilic and exhibited good stability in vitro and specificity for bacteria. Among them, the ^99m^Tc(CO)_3_-labeled complexes showed higher lipophilicity, while ^99m^Tc-labeled and ^99m^TcN-labeled complexes exhibited similar lipophilicity. The biodistribution results in *S. aureus*-infected mice at 120 min postinjection suggested that all complexes had a certain uptake at the abscess site. The biodistribution results in mice suggested that [^99m^Tc]Tc-CN5UBI 29-41 exhibited increased abscess uptake, a more notable target/nontarget ratio and decreased nontarget uptake. In the SPECT imaging study, the site of bacterial infection was clearly observed. [^99m^Tc]Tc-CN5UBI 29-41 shows potential as a bacterial infection imaging probe, justifying further study.

## Data Availability

Data are contained within the article and Appendix A.

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
