# Peer review of "Radiosynthesis and Bioevaluation of 99mTc-Labeled Isocyanide Ubiquicidin 29-41 Derivatives as Potential Agents for Bacterial Infection Imaging"

_ijms, 2024, doi:10.3390/ijms25021045_

Round 1
Reviewer 1 Report
Comments and Suggestions for Authors
The manuscript submitted by Jiang et al. presents a study on the preparation, characterisation and biological evaluation of new 99mTc-labelled Ubiquicidin 29-41 derivatives.
This is a multidisciplinary manuscript covering both chemical and biological techniques. Overall, the study is well thought out.
-The approach to obtaining new derivatives with different chain lengths is justified in the manuscript but it is not clear why the authors choose the 99mTc derivatives for labelling.
-Regarding the methodology, I think the authors are confusing two concepts at the point of the partition coefficient. Log D is the "distribution coefficient" and the partition coefficient would be Log P. The authors talk about Log D and call it a partition coefficient, which is incorrect. The authors should clarify this point.
-The ESI-MS spectra of the compounds are given in the supplementary material and the molecular weight data are given in Table 1 of materials and methods. It would be useful for the manuscript to have the corresponding tables of the molecular formulae proposed for the m/z peaks obtained and other parameters such as mDa Error, DBE... Basically, the complete report of the analysis performed.
Regarding other remarks about the manuscript:
-Scheme 1 should be improved, in terms of image quality and increase the font size.
-In the results of figures 5-7, some statistical analysis is missing. It is recommended that the authors include it in the final version.
-The abstract is difficult to understand and the conclusions could be improved with more information.
Comments on the Quality of English LanguageThe manuscript is well written from a scientific point of view. However, many sentences are too long in the introduction and discussion. The authors should check for such errors and improve them in their final version.
Reviewer 2 Report
Comments and Suggestions for Authors
In the article entitled "Radiosynthesis and Bioevaluation of 99mTc-Labeled Isocyanide Ubiquicidin 29-41 Derivatives as Potential Agents for Bacterial Infection Imaging", the authors added -CN groups with different alkyl chain lengths to the UBI 29-41 peptide (an antimicrobial peptide used in nuclear medicine for the detection of infectious processes) to be labeled with Tc(I)(CO)3, Tc(V)N and Tc(I) cores. While this addition of the -CN group may be original, the possibility that six molecules of UBI-29-41 (MW of 1693 g/mol) can bind around 1 atom of the Tc radiometal with an octahedral geometry exactly as with methoxyisobutisonitrile (MIBI; MW of about 71 g/mol) may be questionable due to steric hindrance. In addition, how is it that in the reversed-phase HPLC radiochromatogram the 99mTc-CN(UBI)6 radio complex (MW greater than 10,158 g/mol considering the 6 UBI molecules) has the lowest and such a different retention time with respect to the carbonyl and nitrile derivatives - what does the presence of two peaks mean? Perhaps the major weakness of the work is the lack of controls, why was only one strain of bacteria used? a negative control, such as fibroblasts or some type of human cells, should have been used to demonstrate specificity towards bacteria. Finally, the biodistribution image in the abscess (again lacking controls) does not correspond to a radiotracer whose abscess/blood uptake ratio is less than 1 over 120 min.
Comments on the Quality of English LanguageNone
Round 2
Reviewer 2 Report
Comments and Suggestions for Authors
Although the authors used a reference related to an evidently unstable radiotracer (99mTc-sulfamethoxazole biodistribution table) to justify their results, I believe that they have generally modified and justified several points questioned in the first review, so that the manuscript could be accepted as is.